# Mississippi Farmers’ Interest in and Experience with Farm to School

**DOI:** 10.3390/ijerph19138025

**Published:** 2022-06-30

**Authors:** Jessica L. Thomson, Tameka I. Walls, Alicia S. Landry

**Affiliations:** 1US Department of Agriculture, Agricultural Research Service, Stoneville, MS 38776, USA; tameka.walls@usda.gov; 2Department of Nutrition and Family Sciences, University of Central Arkansas, Conway, AR 72035, USA; alandry@uca.edu

**Keywords:** farm to school, local foods, Mississippi, online survey, procurement challenges, small farmers

## Abstract

The study’s purpose was to collect demographic and farm characteristics from Mississippi small farmers and to determine their abilities, experiences, and desires to engage in Farm to School (F2S) activities. The online survey was created using items taken from existing F2S surveys. Invitations to participate were sent via email to farmers beginning in October 2021 and ending in January 2022. Descriptive statistics were used to summarize the data. Of the 258 individuals with valid email addresses, 43 (17%) completed the online survey, and 38 fit the definition of small farm (<USD 250,000 in gross cash farm income). Mean farm acreage was 68 (range 1–480 acres). Twenty-six (70%) farms did not have any certifications. Common selling practices included farm stands/stores (n = 18; 49%) and farmers markets (n = 16; 43%). Only 4 farmers (11%) sold to schools with half indicating the experience was difficult. Common challenges included no relationship with school staff (n = 14; 44%) and guarantying quantity/date (n = 11; 34%). Twenty-six (68%) farmers expressed an interest in at least one F2S activity. To facilitate mutually beneficial relationships between small farmers and school food service staff, work is needed to connect the two groups and guide farmers in navigating school procurement rules and regulations.

## 1. Introduction

According to the 2017 Census of Agriculture, 32% of farms in Mississippi, which is the fourth most rural state in the nation, are less than 50 acres in size and 90% of farmers reported yearly sales less than USD 250,000 [1]. While 87% of Mississippi farms are owned by a family or an individual, just 3% (1094 out of 34,988) of farmers sell food directly to consumers with an average value of USD 6959 per farm and less than 1% (n = 201) sell food directly to intermediate markets (e.g., retail markets, institutions, and local food hubs) with an average value of USD 17,980 per farm [1]. Thus, it appears that few farmers in Mississippi are taking advantage of the continuing consumer interest in and demand for locally sourced foods [2] which is unfortunate because direct-to-consumer and intermediate market venues can provide additional income sources for small farmers. Additionally, economists have found that farmers who market goods directly to consumers are more likely to remain in business than those who market only through traditional channels [3].

One market, both direct-to-consumer and intermediate, that has grown exponentially over the last 20 years is Farm to School (F2S) [4,5]. F2S initiatives are designed to connect schools with local or regional producers with the intention of serving locally produced foods in school cafeterias and providing students with food- and agricultural-related educational experiences [4]. Farmers can benefit from F2S participation through market diversification, increased off-season sales, and revenue from surplus or less desirable foods [4]. Additionally, F2S participation can involve more altruistic benefits for farmers including increasing amounts of nutritious foods available to school children and the community and increasing awareness of agricultural practices among students and non-growers [6,7]. A comprehensive review of F2S in the United States was conducted by the United States Department of Agriculture (USDA) Food and Nutrition Service and may be found on their website [4].

In a 2013 study conducted with Mississippi fruit and vegetable growers, only 13% sold to schools although 24% of respondents who had not previously sold to local institutions, such as schools, indicated they were interested in doing so [7]. Growers’ motivation for selling to local institutions included increasing profits, improving community nutrition, and increasing awareness of agricultural practices among students [7]. Additionally, most growers expressed interest in other F2S activities such as having farm visitors and visiting school classrooms [7]. However, several barriers for selling to schools were identified of which the top two were lack of knowledge about how to sell to schools and mismatch between produce availability and academic school year [7]. Thus, past research indicates that small farmers in Mississippi are interested in participating in F2S programs and activities but lack knowledge and experience in doing so. Nonetheless, updated data is needed to determine whether increases in small farmers’ desire to participate and participation in F2S are occurring as well as which barriers to small farmers’ participation in F2S continue to exist.

Mississippi does have an F2S network dedicated to connecting farmers with schools to bring Mississippi-grown products into school cafeterias and F2S is gaining momentum in Mississippi [8]. Yet, more and updated knowledge about small farmers and their farms is needed for F2S to continue to expand and positively affect health outcomes of school children and communities in Mississippi. Therefore, the purpose of this study was to collect updated demographic and farm characteristics data from small farmers in Mississippi and to determine their current abilities, experiences, and desires to engage in F2S activities. As a secondary objective, test–retest reliability of the online survey was conducted.

## 2. Materials and Methods

The Mississippi Farm to School (MS F2S) Study was a two-part online survey that was conducted with farmers (part one) and school food service directors (part two) in Mississippi. Results from the farmer survey are presented in this paper while results from the school food service directors survey will be presented elsewhere. The study was approved and classified as exempt by the Institutional Review Board of Delta State University, Mississippi, USA. All participants provided electronic informed consent prior to completing the survey. Online data collection occurred from October 2021 through March 2022.

Mississippi farmers were identified through a variety of sources including university extension services’ workshop/training attendee lists, USDA Agricultural Research Service (ARS) scientists’ farmer contacts, farmers’ market vendor lists, and Facebook (search terms = Mississippi, farmer, farm, fresh produce, fruits, and vegetables). Farms were classified based on the USDA definition of small farm as an operation with yearly gross cash farm income under USD 250,000 [9]. Farms that fit this definition were classified as small and those that exceeded the income limit were classified as not small.

Prior to conducting the study, a literature review was performed to search for questionnaires that could be used for the present study’s purpose. No single questionnaire was found that covered all the topics of interest; hence, researchers combined and modified items from published questionnaires to create the questionnaires for this study. The majority of farmer survey items were taken from the 2016 Nevada Department of Agriculture F2S Survey and included products raised or produced, harvest months, produce amounts, selling practices including K-12 schools, experience selling to K-12 schools, challenges selling to K-12 schools, and interest in F2S activities [10]. Lists of fruits and vegetables grown were created based on commonly consumed fruits and vegetables [11] and those sold at Mississippi farmers’ markets [12]. The inclusion of value-added products (e.g., homemade jams) was based on the Cornell University F2S Toolkit [13]. The list of farm certifications was based on researcher knowledge and an Internet search. The inclusion of Internet use was based on a USDA Economic Research Service (ERS) report about beginning and experienced farmers’ marketing practices [14]. Farmer demographic characteristics included gender (male or female), age (18–24, 25–44, 45–64, or 65+ years), Hispanic/Latino ethnicity (yes or no), race (black/African American, white, American Indian/Alaska native, Asian, Native Hawaiian/Pacific Islander, other; multiple choices possible), marital status (married, widowed, divorced, separate, never married, or living with partner), and education level (<high school graduate, high school graduate/General Education Development [GED], some college, associate/technical/vocational degree, Bachelor’s degree, Master’s degree, or professional/doctoral degree). Data about harvest months and produce amounts will be presented elsewhere.

The questionnaire (Appendix A) was created using Snap Desktop software, version 12.06 (Snap Surveys Ltd., Portsmouth, NH, USA). Three researchers with experience in F2S methodology or experience working with small farmers reviewed the questionnaire for content and face validity. Additionally, the study team completed the online questionnaire several times to test routing rules and ease of use. Based on the reviews and testing, list revisions and small wording changes were made. The online survey opened in October 2021 and closed in January 2022. Invitations were sent by email using Snap Online (Snap Surveys Ltd., Portsmouth, NH, USA), an online mobile and secure survey management system. The email invitations contained both a link to the questionnaire and a link to opt out of the study. Five reminder emails were sent, approximately 2 weeks apart, to participants who had not yet completed the questionnaire. Additionally, one research team member called individuals whose contact information included a telephone number to encourage them to take the survey. If an individual could not be reached by the fourth call, no further contact attempts were made although messages were left for those with voicemail. Individuals who submitted the questionnaire received a USD 20 e-gift card sent via email.

Approximately two weeks after the survey closed, one research team member conducted follow-up interviews with all participants who agreed to be contacted about their questionnaire responses. Discussion items included fairness of compensation, difficulty completing any questions or series of questions, suggestions for additional information to collect, and methods to contact farmers for future surveys or workshops. Additionally, participants were asked about their willingness to complete another survey to allow for test–retest reliability. The retest survey opened in February 2022 and closed in March 2022. Invitations were again sent by email containing a link to the questionnaire and a link to opt out of the study. Three email reminders were sent, approximately 2 weeks apart, to participants who had not yet completed the questionnaire. Again, participants were called to encourage them to take the survey. Individuals who completed the retest questionnaire received a second USD 20 e-gift card sent via email.

Statistical analyses were performed using SAS^®^ version 9.4 (SAS Institute Inc., Cary, NC, USA). Descriptive statistics including frequencies, percentages, means, and standard deviations were computed. To determine test–retest reliability, simple Cohen’s kappa coefficients (κ) and McNemar’s tests of symmetry were computed for categorical variables with two responses (yes/no) while weighted κ and Bowker’s symmetry tests were computed for categorical variables with more than two responses (Likert-type scales). Additionally, percent agreement was computed for all categorical variables because κ and tests of symmetry do not perform well with sparse data (i.e., zeros in diagonal cells or in both diagonal and off diagonal cells for 2 × 2 contingency tables). Cohen’s kappa coefficients ≥ 0.60 and percent agreement ≥ 80% were considered adequate test–retest reliability criteria [15]. Statistical significance for the tests of symmetry was set at the 0.05 level. For farm size (continuous variable), the intraclass correlation coefficient (ICC) was estimated using a generalized linear mixed model with farm size as the outcome, time as a fixed effect, and participant within time as a random effect using an autoregressive (order 1) covariance structure. An ICC ≥ 0.75 was considered adequate test–retest reliability criterion [16].

## 3. Results

Of the 314 individuals identified as a possible farmer and sent email invitations to participate in the MS F2S Small Farmer Survey, 47 had invalid email addresses, eight indicated they were not farmers, and one was a duplicate email address. Out of the 258 valid survey invitations sent, 215 individuals did not complete the questionnaire (six opened the questionnaire but did not enter any data and 13 opted out) and 43 completed the questionnaire resulting in a 17% response rate. Five of the 43 participants who completed the questionnaire indicated they did not fit the definition of a small farm and hence were excluded from data analysis resulting in an analytic sample size of 38 small farmers.

Participant and farm characteristics are presented in Table 1. Over half (60%) of the participants identified as male, approximately half (49%) were between 45–64 years of age and black (51%), while 68% were married and 46% had a Bachelor’s degree. The majority (70%) of farms did not have any certifications and the most common uses of the Internet were for supplies and equipment (84%), followed by learning resources (66%), although almost half (45%) had a Facebook page for their farm. The most common selling practices included on farm stands/stores (49%) followed by farmers markets (43%) while only four small farmers (11%) sold to schools. Thirty-three (87%) farmers used at least one direct-to-consumer selling method (on/off farm stand/store, farmers’ market, CSA, roadside, u-pick, or sale barn/stockyard) while 12 (32%) used at least one intermediate marketing venue (commercial vendor, restaurant, school, distributor, or cooperative). Farms ranged in size from 1 to 480 acres with a mean size of 68 acres (Table 1).

Farm products are presented in Table 2. Approximately two-thirds (68%) of the farmers grew at least one type of fruit of which the most common were cantaloupe (27%), watermelon (24%), blueberries (22%), and strawberries (22%). Approximately three-fourths of the farmers grew at least one type of vegetable of which the most common were okra (57%), cucumbers (54%), greens (54%), and yellow squash (54%). The mean number of types of fruits and vegetables grown were 2 and 7, respectively. Approximately three-fourths (73%) of the farmers grew or raised products other than fruits and vegetables with the most common including herbs (27%), beef (24%), and eggs (22%). Additionally, 32% of the farmers produced canned goods. Only five farmers did not grow fruits or vegetables, and of those five, two did not produce some type of food for human consumption. One farmer grew trees, and the other farmer was establishing their operation.

The farmers’ F2S experience and interest are presented in Table 3. Of the 4 farmers who sold to schools, 2 (50%) indicated their selling experience was difficult. The most common challenges reported for selling to K-12 schools included no relationship with school staff (44%) followed by guarantying quantity or date (34%). No farmer reported that “price was too high” as a challenge. Strong interest in F2S activities (as indicated by strongly agree responses) ranged from 48% for farm visitors to 26% for forward contracting. Twenty-six (68%) farmers expressed an interest in at least one F2S activity (mean = 2.4, SD = 2.0).

Of the 38 small farmers who completed the initial (test) survey, 29 (76%) agreed to be contacted about their survey responses. All 29 participants were contacted for follow-up interviews and invited to take the survey a second time; 25 (86%) completed the retest survey. The mean time between test and retest surveys was 83 days (SD = 27 days; range = 35–130 days). Test–retest statistics for participant and farm characteristics are presented in Table 1. Participant demographic characteristics exhibited adequate test–retest reliability (κ ≥ 0.82 and percent agreement ≥ 91%). Farm characteristics exhibited adequate reliability for three of six farm certification variables, three of nine internet usage variables, and four of nine selling practice variables (0.61 ≤ κ ≤ 1.00). Of the 14 variables with inadequate reliability, three (κ ≤ 0) were due to sparse data. All of the variables exhibited adequate percent agreement (80% to 100%) except three internet usages and two selling practices (58% to 75%). None of the tests of symmetry were significant indicating that participants were as likely to respond affirmatively at test and negatively at retest than vice versa. Farm size also exhibited adequate test–retest reliability (ICC = 0.92).

Test–retest statistics for farm products are presented in Table 2. Seven of the 14 fruits, 15 of the 28 vegetables, two of the nine other products, and three of the four home-made goods exhibited adequate test–retest reliability (0.63 ≤ κ ≤ 1.00). Of the 28 variables with inadequate reliability, seven (κ ≤ 0) were due to sparse data. All 14 fruits, 21 of the 28 vegetables, seven of the nine other products, and all four of the home-made goods exhibited adequate percent agreement (80% to 100%). Only the symmetry test for pears was significant indicating participants were more likely to select pears on the test questionnaire and not select it on the retest questionnaire than vice versa.

Test–retest statistics for the farmers’ F2S experience and interest are presented in Table 3. Selling and experience selling to K-12 schools and two school selling challenges had sufficient test–retest reliability (0.61 ≤ κ ≤ 1.00). Of the seven variables with inadequate reliability, one was due to sparse data (κ ≤ 0). Selling to K-12 schools and six of the 11 school selling challenges had adequate percent agreement (83% to 100%). None of the five F2S interest variables had sufficient test–retest reliability nor percent agreement. No test of symmetry was significant for any of the F2S experience or interest variables.

Of the 29 small farmers who agreed to be contacted about their survey responses, 27 (93%) were reached for follow-up interviews. Twenty-five (93%) farmers indicated that compensation for completing the survey was fair whereas one farmer indicated compensation was not necessary and the other suggested a higher amount. Twenty-five farmers (93%) also indicated that the survey was not difficult to complete whereas one farmer indicated difficulty combining information for four farm locations and the other indicated the survey was too long. Only one farmer had suggestions for additional information to collect—number of farm employees and ability to provide processed produce (e.g., chopped) to schools. Suggested methods to contact farmers for future surveys or workshops included: emails; text messages; telephone calls and voice messages; USDA Farm Service Agency and Natural Resources Conservation Service lists; university extension services lists; face-to-face via farm/agricultural conferences, associations, and cooperatives; social media; and word of mouth.

## 4. Discussion

This study provides updated information about farm characteristics and products in conjunction with current F2S activities and interest from small farmers in Mississippi. Although not designed to be representative of all small farmers in Mississippi, some farmer characteristics in the current study were remarkably similar to Mississippi producer characteristics reported in the 2017 Census of Agriculture. For Mississippi producers in the 2017 Census who reported <USD 250,000 in market value of products sold, 66% were male, 45% were between 45 and 64 years of age, and 1% were Hispanic [1] as compared to 60% male, 49% between 45 and 64 years of age, and 3% Hispanic in the current study. However, only 14% of Mississippi producers were African American and 64% reported access to the Internet in the 2017 Census [1] as compared to 51% African American and 100% reported access to the Internet in the current study. Thus, although the sample of farmers in the current study is not representative of all farmers in Mississippi, characteristics are similar to those from the 2017 Census of Agriculture sample of Mississippi farmers.

In the current study, only 30% of farmers indicated they had some type of farm certification (e.g., GAP, GHP) which is similar to the relatively low level (28%) reported in a 2013 survey of fruit and vegetable growers in Mississippi [7]. Additionally, similar to the 2013 Mississippi study [7], more farmers in the current study reported selling their products using direct-to-consumer methods versus intermediate marketing venues, such as K-12 schools. Reasons for the higher frequency of direct-to-consumer selling methods observed in the current study may involve challenges unique to small farmers when selling to intermediate markets. These challenges include difficulty maintaining a consistent supply of local food and food safety and management standards that are often required by retailers [17]. For example, farmers must have GAP/GHP certification if they wish to supply produce to the USDA Department of Defense Fresh Fruit and Vegetable Program (DoD Fresh), a purchasing program designed to encourage serving locally grown or locally produced agricultural products in school meals [18]. Additionally, while farmers are not required to have GAP/GHP certification to sell their products to schools, schools may require it. Although GAP/GHP audit costs can be prohibitively high for small farmers, the Mississippi Department of Agriculture and Commerce (MDAC) has a cost-share program for Mississippi farmers to provide financial assistance to cover the cost of certification [19]. Further, the Mississippi F2S Network has a resource webpage dedicated to helping farmers connect with schools, including a food safety checklist with production practices, product handling, transportation, facilities, and worker health and hygiene items that are covered in GAP/GHP audits [20].

Few small farmers in the current study had experience selling to K-12 schools and almost half indicated they did not have relationships with school staff, although close to 70% expressed an interest in participating in at least one F2S activity. These results are similar to those from a Michigan study for which half of the farmers surveyed expressed an interest in selling to institutions such as K-12 schools, but only 7% had experience selling to institutions [21]. In another study conducted with small farmers in South Carolina, 85% expressed an interest in selling to K-12 schools although none had experience selling to schools and challenges included no relationship with school staff [22]. Encouragingly, 95% of the small farmers in the current study grew or raised food for human consumption with the majority (87%) growing fruits or vegetables, the top two food categories sourced locally by Mississippi public school districts [23]. Taken together, these results suggest that mechanisms are needed to connect farmers with school staff to facilitate mutually beneficial partnerships between the two groups that will result in more locally sourced foods being served in school cafeterias.

For farmers to connect with schools, they need to be visible to or discoverable by school food service staff. For both farmers and school food service staff interested in F2S, the MDAC advocates for the use of Mississippi Market Maker, a free internet marketing tool, provided by the Mississippi State University Extension Service (MSU-ES), that links growers and producers with consumers [18,24,25]. Yet, of the 258 farms listed on the website, not one was a farmer who participated in the current study. In addition, the Mississippi F2S Network resource webpage for farmers contains advice on how and when to connect with schools [20]. Hence, there are resources available to help small farmers connect with schools and participate in F2S programs. However, for these resources to be useful, farmers must be aware of their existence. One possible avenue to accomplish this awareness is through state university extension service that offer technical advice and services to support farmers in their agricultural production. Local Flavor, a MSU-ES initiative, brings Extension faculty and agents together to support the development of Mississippi’s local foods industry by connecting individuals searching for information about food safety, agricultural production, business development, economics, and policy with resources across the state [26].

Test–retest reliability (κ statistic) was less than optimal for the majority (53%) of questionnaire items. Reasons for low reliability include sparse data, small sample sizes, and long periods between completion of the initial and retest questionnaires. The mean test–retest interval was six weeks in the present study which is three times greater than the most frequently recommended test–retest target interval of two weeks [27]. Additionally, the κ statistic may have excessively lowered the level of agreement for many questionnaire items because, although it corrects for chance agreement, it is not likely that the farmers were guessing when providing their responses [15]. Based on percent agreement, test–retest reliability was sufficient for approximately three-fourths (76%) of the questionnaire items. Another possible reason for the low reliability observed is that some questionnaire items simply may not be very stable by their nature. Initial questionnaires were completed in late fall and early winter when little planting takes place, and farmers may have responded based on produce grown in the previous spring or summer. Retest questionnaires were completed in late winter to early spring when planting may have just begun, and produce may have differed from the previous year’s planting. Additionally, interest in F2S activities may be high during winter months when farmers have more “down time” with interest dwindling during spring and summer when planting and harvesting are occurring, and “free time” is limited.

Several limitations of this study bear mentioning. The most current USDA definition of small family farm [28] was not used, sample size was small and not representative of all small farmers in Mississippi, and parameters for determining test–retest reliability were not optimal. To address these limitations and improve recruitment of the target population, future iterations of this survey will incorporate the following modifications: use of most recent USDA definition for small family farms; inclusion of definitions for farm certifications (e.g., Good Agricultural Practices includes food safety); addition of more social media choices to Internet use item; addition of sale barn/stockyard, u-pick, and website/Internet choices for selling practices; replacement of “price too high” with “low profitability” and addition of “lack of skilled labor for food preparation” choices for school selling challenges; and replacement of 5-point Likert-type agreement scale with three choices—no, maybe, yes—for F2S interest item. Additionally, test–retest reliability will be repeated to include a shorter time interval between survey administrations. Importantly, a more comprehensive list of Mississippi small farmers will be created using sources such as the USDA Farm Service Agency, USDA Natural Resources Conservation Service, and community organizations (e.g., Mississippi F2S Network, Mississippi Association of Cooperatives, and Mississippi Fruit and Vegetable Growers Association).

## 5. Conclusions

This study contributes valuable and updated findings about Mississippi small farmers experience with and interest in F2S activities. Based on the study findings, most farmers lacked experience selling their products to K-12 schools although the majority expressed interest in doing so. However, to facilitate successful and mutually beneficial relationships between small farmers and K-12 school food service staff much work is needed to connect the two groups and to educate and guide farmers in navigating the rules and regulations surrounding local procurement in schools, including obtaining farm certifications.

USDA’s recently proposed framework for shoring up the food supply chain and transforming the food system [29] may help mitigate some of the barriers small farmers encounter when attempting to participate in F2S. Specifically, continued support for the Food Safety Certification for Specialty Crops Program will lessen costs small farmers incur for GAP/GHP certification. Additionally, investment in the creation of regional food business centers that will provide technical assistance and capacity building support to small farmers may help alleviate challenges such as consistency in produce quantity, storage space, and delivery logistics/transportation experienced by small farmers. Finally, grant support for leveraging increased commodity purchases through F2S may encourage school food service staff to connect with small farmers to increase procurement and use of local foods in school meals. Collecting longitudinal data will be key for determining whether the Food System Transformation initiatives positively impact small farmers’ participation in F2S and ultimately, their long-term economic viability.

## Figures and Tables

**Table 1 ijerph-19-08025-t001:** Participant and Farm Characteristics with Test–Retest Statistics ^a^: Mississippi Farm to School Small Farmer Survey 2021–2022.

	Original Survey ^b^		Test–Retest ^c^		
Characteristic	n	%	κ	%A	MP
Gender			**1.00**	**100.0**	^d^
Male	22	59.5			
Female	15	40.5			
Age (years)			**1.00**	**100.0**	^d^
25–44	11	29.7			
45–64	18	48.6			
≥65	8	21.6			
Hispanic or Latino	1	2.7	**1.00**	**100.0**	^d^
Race (multiple choice)					
Black or African American	19	51.4	**0.82**	**91.3**	0.157
White	18	48.6	**1.00**	**100.0**	^d^
American Indian or Alaska Native	2	5.4	**1.00**	**100.0**	^d^
Marital Status			**0.97**	**95.5**	<0.999
Married or living with partner	25	67.6			
Widowed or divorced	7	18.9			
Never married	5	13.5			
Education Level			**1.00**	**100.0**	^d^
≤High school graduate/GED	3	8.1			
Some college ^e^	5	13.5			
Bachelor’s degree	17	45.9			
Master’s degree	6	16.2			
Professional or doctoral degree	4	10.8			
Farm Certifications					
Good Agricultural Practices (GAP)	6	16.2	0.50	**88.0**	0.564
Good Handling Practices (GHP)	4	10.8	**0.62**	**92.0**	<0.999
Naturally grown	3	8.1	0.25	**84.0**	0.317
Non-GMO	1	2.7	^f^	**100.0**	^d^
Organic	1	2.7	**1.00**	**100.0**	^d^
None	26	70.3	0.56	**80.0**	0.655
Internet Usage					
Supplies and equipment	32	84.2	0.17	**80.0**	0.655
Learning resources	25	65.8	0.34	72.0	0.257
Price and market information	20	52.6	0.37	68.0	0.157
Grants and subsidies	18	47.4	**0.61**	**80.0**	0.180
Farm Facebook page	17	44.7	**0.76**	**88.0**	0.564
Farm website	12	31.6	**0.82**	**92.0**	<0.999
Products and services	11	28.9	0.45	76.0	0.414
Other ^g^	1	2.6	0.00	**96.0**	0.317
None	1	2.6	0.00	**96.0**	0.317
Selling Practices					
On farm stand/store	18	48.6	0.14	58.3	0.206
Farmers’ market	16	43.2	**0.75**	**87.5**	0.564
Community supported agriculture	9	24.3	**0.90**	**95.8**	0.317
Off farm stand/store	8	21.6	0.49	**83.3**	<0.999
Commercial vendor	6	16.2	0.51	**87.5**	0.083
Restaurant	6	16.2	**0.88**	**95.8**	0.317
School	4	10.8	**0.86**	**95.8**	0.317
Distributor	1	2.7	−0.04	**91.7**	<0.999
Other ^h^	8	21.6	0.47	75.0	0.414
	**Original Survey**				**Retest**
	**Mean**	**Median**	**SD**	**Range**	**ICC**
Farm size (acres)	68	24	101	1–480	**0.92**

κ, Cohen’s kappa coefficient; %A, percent agreement; MP, McNemar’s test *p*-value; GED, General Education Development; GMO, genetically modified organism; SD, standard deviation; ICC, intraclass correlation coefficient. ^a^ Test (original) sample size = 38; retest sample size = 25. ^b^ Sample sizes may not add to 38 due to missing data. ^c^ Bolded values indicate adequate test–retest reliability. ^d^ McNemar’s test not computed due to 100% agreement (off-diagonal cell counts = 0). ^e^ Included Associate, technical or vocational degree. ^f^ κ not computed because 1 participant who selected non-GMO did not participate in retest survey (all responses were zero). ^g^ Marketing. ^h^ Cooperative, roadside, sale barn/stockyard, u-pick, website, word of mouth.

**Table 2 ijerph-19-08025-t002:** Farm Products with Test–Retest Statistics ^a^: Mississippi Farm to School Small Farmer Survey 2021–2022.

	Original Survey ^b^		Test–Retest ^c^		
Food Type	n	%	κ	%A	MP
Fruits					
Apples	1	2.7	**1.00**	**100.0**	^d^
Blackberries	6	16.2	**0.86**	**96.0**	0.317
Blueberries	8	21.6	**0.78**	**92.0**	<0.999
Cantaloupe	10	27.0	0.42	**80.0**	0.655
Figs	4	10.8	−0.04	**92.0**	<0.999
Grapes muscadine	5	13.5	**0.65**	**96.0**	0.317
Peaches	2	5.4	0.47	**92.0**	0.157
Pears	7	18.9	0.23	**80.0**	0.025
Plums	4	10.8	0.46	**92.0**	<0.999
Raspberries	1	2.7	0.00	**96.0**	0.317
Strawberries	8	21.6	0.32	**80.0**	0.655
Watermelon	9	24.3	**0.68**	**84.0**	0.317
Other ^e^	2	5.4	**0.63**	**92.0**	0.157
None	12	32.4	**0.69**	**88.0**	0.564
Vegetables					
Bell peppers	16	45.7	**0.83**	**91.3**	<0.999
Black eyed peas	3	8.6	0.33	**87.0**	0.564
Broccoli	10	28.6	0.35	69.6	0.706
Butter beans	12	34.3	0.43	73.9	0.414
Cabbage	13	37.1	**0.72**	**87.0**	0.564
Carrots	9	25.7	0.54	78.3	0.180
Cauliflower	7	20.0	**0.77**	**91.3**	<0.999
Cucumbers	19	54.3	**0.74**	**87.0**	0.564
Eggplant	11	31.4	**1.00**	**100.0**	^d^
Green beans	8	22.9	0.40	78.3	0.655
Green peas	3	8.6	0.10	73.9	0.414
Greens	19	54.3	0.56	78.3	0.655
Kale	11	31.4	**0.82**	**91.3**	0.157
Kohlrabi	3	8.6	**0.83**	**95.7**	0.317
Leeks	2	5.7	**1.00**	**100.0**	^d^
Mushrooms	1	2.9	**1.00**	**100.0**	^d^
Okra	20	57.1	**0.73**	**87.0**	0.564
Onions	7	20.0	**0.77**	**91.3**	<0.999
Purple hull peas	18	51.4	0.57	78.3	0.655
Radishes	8	22.9	0.55	**82.6**	<0.999
Sweet corn	10	28.6	0.55	**82.6**	<0.999
Sweet potatoes	1	2.9	−0.06	**87.0**	0.564
Swiss chard	5	14.3	**0.88**	**95.7**	0.317
Tomatoes	18	51.4	**0.74**	**87.0**	0.564
White potatoes	4	11.4	0.50	**87.0**	0.564
Yellow squash	19	54.3	**0.82**	**91.3**	<0.999
Zucchini	15	42.9	**0.91**	**95.7**	0.317
Other ^f^	4	11.4	0.33	**87.0**	0.564
None	9	25.7	**0.68**	**87.0**	0.564
Other					
Beef	9	24.3	**0.78**	**91.3**	0.157
Chicken	3	8.1	**0.65**	**95.7**	0.317
Eggs	8	21.6	0.33	**87.0**	0.564
Herbs	10	27.0	0.43	73.9	<0.999
Honey	3	8.1	0.50	**87.0**	0.564
Pecans	7	18.9	−0.07	**82.6**	0.317
Pork	1	2.7	0.00	**95.7**	0.317
Other ^g^	3	8.1	−0.05	**91.3**	<0.999
None	10	27.0	−0.18	52.2	0.763
Home-Made					
Canned goods	12	32.4	**1.00**	**100.0**	^d^
Baked goods	2	5.4	**0.65**	**95.8**	0.317
Other ^h^	3	8.1	0.47	**91.7**	0.157
None	24	64.9	**0.90**	**95.8**	0.317
**Original Survey**					
**Produce**	**Mean**	**Median**	**SD**	**Range**	
Fruits	1.8	2.0	1.6	0–5	
Vegetables	7.2	7.0	7.4	0–23	

κ, Kappa coefficient; %A, percent agreement; MP, McNemar’s test *p*-value. ^a^ Test (original) sample size = 38; retest sample size = 25. ^b^ Sample sizes may not add to 38 due to missing data. ^c^ Bolded values indicate adequate test–retest reliability. ^d^ McNemar’s test not computed due to 100% agreement (off-diagonal cell counts = 0). ^e^ Japanese persimmons, pumpkins. ^f^ Butternut squash, lettuce, pinkeye peas, banana peppers. ^g^ Goats, catfish, hemp. ^h^ Pesto, pickles, popsicles.

**Table 3 ijerph-19-08025-t003:** Farm to School Experience and Interest with Test–Retest Statistics ^a^: Mississippi Farm to School Small Farmer Survey 2021–2022.

	**Original Survey ^b^**		**Test–Retest ^c^**							
**Measure**	**n**	**%**	**κ**	**%A**	**MP**					
Sold to K-12 Schools	4	10.8	**1.00**	**100.0**	NA					
K-12 Selling Experience			**0.67**	50.0	0.801					
Very easy	0	0.0								
Easy	1	25.0								
Neither easy nor difficult	1	25.0								
Difficult	2	50.0								
Very difficult	0	0.0								
K-12 Selling Challenges										
No relationship with staff	14	43.8	0.31	66.7	0.414					
Guarantying quantity/date	11	34.4	**0.61**	**83.3**	0.564					
Insurance liability	8	25.0	0.45	77.8	0.317					
Food safety regulations	7	21.9	0.51	77.8	0.317					
Product seasonality	6	18.8	−0.08	**83.3**	0.564					
Storage space	6	18.8	0.37	77.8	0.317					
Delivery logistics/transportation	4	12.5	0.31	**83.3**	0.564					
Schools not interested	1	3.1	**1.00**	**100.0**	^d^					
Limiting school contract	1	3.1	^e^	**100.0**	^d^					
Price too high	0	0.0	^e^	**100.0**	^d^					
Other ^f^	12	37.5	0.45	77.8	0.317					
	**Original Survey ^b^**									
	**Strongly Disagree**		**Somewhat Disagree**		**Neither**		**Somewhat Agree**		**Strongly Agree**	
**Farm to School Interest**	**n**	**%**	**n**	**%**	**n**	**%**	**n**	**%**	**n**	**%**
Farm visitors	5	16.1	1	3.2	3	9.7	8	25.8	15	48.4
Visiting schools	7	22.6	0	0.0	3	9.7	9	29.0	13	41.9
Planning next growing season	8	25.8	1	3.2	5	16.1	8	25.8	10	32.3
Plant more acres for schools	10	32.3	3	9.7	7	22.6	3	9.7	10	32.3
Forward contracting	9	29.0	1	3.2	6	19.4	7	22.6	8	25.8
	**Test–Retest ^c^**									
	**κ**	**%A**	**BP**							
Farm visitors	0.13	57.1	0.973							
Visiting schools	0.02	47.6	0.596							
Planning next growing season	0.22	42.1	0.441							
Plant more acres for schools	0.23	44.4	0.532							
Forward contracting	0.17	38.9	0.423							

κ, Cohen’s kappa coefficient; %A, percent agreement; MP, McNemar’s test *p*-value; BP, Bowker’s test *p*-value. ^a^ Test (original) sample size = 38; retest sample size = 25. ^b^ Sample sizes may not add to 38 due to missing data. ^c^ Bolded values indicate adequate test–retest reliability. ^d^ McNemar’s test not computed due to 100% agreement (off-diagonal cell counts = 0). ^e^ κ not computed because 1 participant who selected school contract did not participate in retest survey or all responses were zero. ^f^ Do not need product (trees), donate (do not sell) produce, scale of production, u-pick, never tried.

## Data Availability

The data presented in this study will be made openly available after publication in the USDA National Agricultural Library’s Ag Data Commons.

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
