# Peer review of "Mississippi Farmers’ Interest in and Experience with Farm to School"

_ijerph, 2022, doi:10.3390/ijerph19138025_

Round 1

Reviewer 1 Report

The authors used an online survey to collect demographic and farm characteristics from Mississippi small farmers. The aim of the study was to determine  abilities, experiences, and desires of surveyed farms to engage in Farm to School (F2S) activities.

The manuscript is well written but the authors devote too much attention to the (descriptive) statistical analysis and too little attention to the interpretation of the survey findings.  The result is that the manuscript looks more as a “technical report” than a “scientific paper”. I recommend that authors rewrite the paper and submit again for a new evaluation.

The authors should pay more attention to investigate in what way the farm characteristics influence  the intention to engage in Farm to School (F2S) activities and what can be done to solve the problems that prevent a larger participation.

The authors focus on the statistical analysis and give a very detailed description of the phenomenon, but they devote just a few, sparse words to the interpretation of results.

In other words, the authors make a very good description of the phenomenon, but they do not explain what are the problems highlighted by the picture and do not suggest ways to solve them and improve the current situation.

As a reader I would like to know
-Why surveyed farmers prefer direct to intermediate selling?
-What drives the attitude toward direct or intermediate selling?
-Are the rules and regulations surrounding local procurement in schools hidden costs for farmers? can be changed or maybe farmers may join to be able to spread these costs?
-What mechanisms can be used to connect farmers to schools?

I also attach a copy of the manuscript with my comments to the discussion section.

Author Response

We thank the reviewers for their thoughtful and thorough review of our manuscript. We have incorporated many of the suggestions provided and believe that these revisions have improved the comprehensive nature and clarity of our manuscript. As is often the case, we had to balance incorporating reviewers’ suggestions with creating a comprehensive yet concise manuscript that will attract and hold readers’ attention and interest. We hope our responses and manuscript edits (made using track changes) have adequately addressed the reviewers’ concerns. Our responses to the comments and suggestions follow.

  1. The authors used an online survey to collect demographic and farm characteristics from Mississippi small farmers. The aim of the study was to determine abilities, experiences, and desires of surveyed farms to engage in Farm to School (F2S) activities. The manuscript is well written but the authors devote too much attention to the (descriptive) statistical analysis and too little attention to the interpretation of the survey findings. The result is that the manuscript looks more as a “technical report” than a “scientific paper”. I recommend that authors rewrite the paper and submit again for a new evaluation.

Response: We thank the Reviewer for recognizing merit in our manuscript as well as identifying areas for improvement. We have responded to the Reviewer’s questions and comments following.

  1. The authors should pay more attention to investigate in what way the farm characteristics influence the intention to engage in Farm to School (F2S) activities and what can be done to solve the problems that prevent a larger participation.

Response: We agree that an analysis exploring farm characteristics associated with F2S interest/activities would be of interest. However, the size of our sample limits our ability to conduct an analysis that would provide meaningful results worthy of reporting. We have added information to the Discussion to address barriers to farmers’ participation in F2S (lines 288-294, 310-324).

  1. The authors focus on the statistical analysis and give a very detailed description of the phenomenon, but they devote just a few, sparse words to the interpretation of results. In other words, the authors make a very good description of the phenomenon, but they do not explain what are the problems highlighted by the picture and do not suggest ways to solve them and improve the current situation.

Response: The Reviewer makes a valid point. We have expanded the Discussion to include suggestions for ameliorating some of the challenges identified by farmers in our study (lines 288-294, 310-324).

  1. As a reader I would like to know why surveyed farmers prefer direct to intermediate selling?

Response: The Reviewer raises an interesting question. We did not ask the farmers in our study their reasons for selling products to specific markets because our focus was on F2S. However, using previous research, we have added possible reasons why small farmers may prefer direct-to-consumer selling methods to the Discussion (lines 279-283).

  1. As a reader I would like to know what drives the attitude toward direct or intermediate selling?

Response: Again, we did not ask the farmers in our study their reasons for selling their products to specific markets because our focus was on F2S. Using previous research, we have added possible reasons small farmers may prefer direct-to-consumer selling methods to the Discussion (lines 279-283).

  1. As a reader I would like to know are the rules and regulations surrounding local procurement in schools hidden costs for farmers? can be changed or maybe farmers may join to be able to spread these costs?

Response: We have added information regarding selling farm products to schools including resources for farmers to the Discussion (lines 288-294, 310-317). To our knowledge, there are no “hidden costs” associated with farmers selling to schools.

  1. As a reader I would like to know what mechanisms can be used to connect farmers to schools?

Response: The Reviewer raises an excellent question. We have added suggestion for making these connections to the Discussion (lines 310-324).

Reviewer 2 Report

Comments IJERPH 1763693

Abstract

Abstract needs to be rewritten. It is proposed that the abstract be described as follows: 1) Introduce the research problem. 2) Identify the main objectives and scope of the research. 3) Mention the materials and methods used. 4) Meaningful results (numerical, percentages); and 5) Main conclusions (contributions).

Introduction

The introduction is too short and focused more on the F2S and sales of small farmers in Mississippi. The introduction should be dedicated to introducing this topic clearly, presenting the background, highlighting the relevant literature and making it clear why examining this topic is of value. Also, no great deal was done in identifying the research gap which this study seeks to fill. As such, I suggest the introduction requires re-writing to more closely match the title and topic of this paper. Also the objectives of the paper should be stated in the Introduction.

The authors should create a Section on Literature Review after the Introduction. The gap or shortcomings of the literature should be stated.

Materials and Methods

Please where is Delta State University that its Institutional Review Board approved and classified this study as exempt? Please indicate the country and if possible the full address of the University.

Include copy(ies) of the questionnaire(s) used for this study as supplementary materials or appendix.

Results

Thirty eight (38) is too small. Sample size appears small for generalization especially at the policy front. Authors should think about this.

Please calculate the mean age of the farmers and presented same in Table?

Discussion

The discussion of the results needs to be improved. Currently, it is unfocused. In addition, much of the discussion is reiterating the results, while there is only limited discussion of the findings in light of previous and related literature. The discussion section overall would be improved by a much clearer focus on elucidating key findings in light of existing literature and how this study builds on/relates to past findings. Specifically, there needs to be significantly more references and linkage to relevant literature to back up many of the points made in this section.

Conclusion

It is better to offer decision-makers suggestions in the conclusion section. Also, provide solutions. I suggest authors should add some recommendations and policy implications for this study derived from the findings. Please note that recommendations must be ACTIONABLE.

References

Since this is not Bibliography, authors should cross check to ensure that only authors cited in the body of literature in the paper appear in the Reference section

Author Response

We thank the reviewers for their thoughtful and thorough review of our manuscript. We have incorporated many of the suggestions provided and believe that these revisions have improved the comprehensive nature and clarity of our manuscript. As is often the case, we had to balance incorporating reviewers’ suggestions with creating a comprehensive yet concise manuscript that will attract and hold readers’ attention and interest. We hope our responses and manuscript edits (made using track changes) have adequately addressed the reviewers’ concerns. Our responses to the comments and suggestions follow.

  1. Abstract: Abstract needs to be rewritten. It is proposed that the abstract be described as follows: 1) Introduce the research problem. 2) Identify the main objectives and scope of the research. 3) Mention the materials and methods used. 4) Meaningful results (numerical, percentages); and 5) Main conclusions (contributions).

Response: Thank you for your comment. We have followed the journal’s guidelines for abstracts that include stating the purpose of the study (collect farm/farmer data related to F2S), describing the main methods (online survey), summarizing the main findings (farm characteristics, selling practices and F2S activities and challenges with numerical percentages), and stating the main conclusions without exaggerating them (work is needed to connect farmers to schools). Given the journal’s limit of 200 words for the abstract, we had to be concise in writing the abstract.

  1. Introduction: The introduction is too short and focused more on the F2S and sales of small farmers in Mississippi. The introduction should be dedicated to introducing this topic clearly, presenting the background, highlighting the relevant literature and making it clear why examining this topic is of value. Also, no great deal was done in identifying the research gap which this study seeks to fill. As such, I suggest the introduction requires re-writing to more closely match the title and topic of this paper. Also the objectives of the paper should be stated in the Introduction.

Response: The first paragraph of the Introduction introduces our target population (small farmers in Mississippi) while the second paragraph explains potential benefits of F2S participation for small farmers. However, we do agree that we did not include past research regarding small farmers interest in and experience with F2S. Hence, we have expanded the Introduction to include this information (lines 66-76). We also have more clearly identified how our work fills a research gap (lines 76-78). We stated the objectives of the paper in the last 2 sentences of the Introduction.

  1. The authors should create a Section on Literature Review after the Introduction. The gap or shortcomings of the literature should be stated.

Response: Thank you for this suggestion. We have expanded the Introduction to include past research regarding small farmers interest in and experience with F2S (lines 66-76). We also have more clearly identified the research gap filled by our study (lines 76-78).

  1. Materials and Methods: Please where is Delta State University that its Institutional Review Board approved and classified this study as exempt? Please indicate the country and if possible the full address of the University.

Response: We have checked 5 recent publications in IJERPH that report data from human studies. All 5 included the name of the IRB institution that approved their study but not the country or the address for the IRB institution. Additionally, the journal’s instructions for authors state that the name of the ethics committee or institution should be given and do not require country or address to be given. We defer this decision to the Editor and will provide the information as required.

  1. Include copy(ies) of the questionnaire(s) used for this study as supplementary materials or appendix.

Response: This is an excellent suggestion. We have included the questionnaire as supplementary material (line 91).

  1. Results: Thirty eight (38) is too small. Sample size appears small for generalization especially at the policy front. Authors should think about this.

Response: We agree that our sample size is small and did list it as a limitation of our study in the last paragraph of the Discussion. We also have not attempted to generalize our study results to represent all small farmers in Mississippi.

  1. Please calculate the mean age of the farmers and presented same in Table?

Response: We can appreciate the Reviewer’s interest in the farmers’ mean age. It is not possible for us to calculate a mean age because the survey collected age as categories. We have included the demographic characteristics collected to clarify this in the Methods (lines 85-92).

  1. Discussion: The discussion of the results needs to be improved. Currently, it is unfocused. In addition, much of the discussion is reiterating the results, while there is only limited discussion of the findings in light of previous and related literature. The discussion section overall would be improved by a much clearer focus on elucidating key findings in light of existing literature and how this study builds on/relates to past findings. Specifically, there needs to be significantly more references and linkage to relevant literature to back up many of the points made in this section.

Response: We can appreciate the Reviewer’s suggestions to improve our Discussion. Hence, we have substantively revised the Discussion and hope these revisions address the Reviewer’s concerns (lines 265-280, 296-310).

  1. Conclusion: It is better to offer decision-makers suggestions in the conclusion section. Also, provide solutions. I suggest authors should add some recommendations and policy implications for this study derived from the findings. Please note that recommendations must be ACTIONABLE.

Response: This is an excellent suggestion. We have added a paragraph to the Conclusion that discusses recent initiatives that may help overcome some of the barriers of F2S participation experienced by small farmers (lines 370-381).

  1. References: Since this is not Bibliography, authors should cross check to ensure that only authors cited in the body of literature in the paper appear in the Reference section.

Response: We thank the Reviewer for this section and have ensured that only references cited in the manuscript appear in the reference list.

Reviewer 3 Report

(1)    Abstract requires improvements with respect to sentence styles (percentage and numbers). Moreover, policy issues should come clearly in the abstract itself.

(2)    In line 72 and 73, the paper says “Prior to conducting the study, a literature review was performed to search for questionnaires that could be used for the present study’s purpose.”. But there is no review of relevant literature on the topic. The theoretical justification of the study objective should be given.

(3)    Results and discussions (section 3 and 4) should be improved.

(4)     Formats in data Tables are not looking good. Reformats should be done.  

(5)    Conclusion in section 5 is also not strong. Please give a good summary and proper policy implications.  

(6)    It is known to all that every study suffers from some limitations, like data problem, model estimation/specification problem etc. But authors have not mentioned such limitations at the concluding section.

The paper should be revised in the light of above comments and suggestions before accepting for publication in the journal.

Author Response

We thank the reviewers for their thoughtful and thorough review of our manuscript. We have incorporated many of the suggestions provided and believe that these revisions have improved the comprehensive nature and clarity of our manuscript. As is often the case, we had to balance incorporating reviewers’ suggestions with creating a comprehensive yet concise manuscript that will attract and hold readers’ attention and interest. We hope our responses and manuscript edits (made using track changes) have adequately addressed the reviewers’ concerns. Our responses to the comments and suggestions follow.

  1. Abstract requires improvements with respect to sentence styles (percentage and numbers). Moreover, policy issues should come clearly in the abstract itself.

Response: We would be happy to revise the abstract if provided with specific improvements as we followed the journal’s instructions when writing the abstract. Given the journal’s limit of 200 words for the abstract, we had to be concise in our writing and do not have the space to add policy issues to the abstract without the Editor’s permission.

  1. In line 72 and 73, the paper says “Prior to conducting the study, a literature review was performed to search for questionnaires that could be used for the present study’s purpose.”. But there is no review of relevant literature on the topic. The theoretical justification of the study objective should be given.

Response: We have expanded the Introduction to include past research regarding small farmers interest in and experience with F2S (lines 66-76). We also have more clearly identified how our work fills a research gap (lines 76-78).

  1. Results and discussions (section 3 and 4) should be improved.

Response: Thank you for expressing this concern. If the Reviewer could provide specific examples of suggested revisions for the Results section, we could improve this section accordingly. The Discussion has been substantively revised based upon the other reviewers’ suggestions.

  1. Formats in data Tables are not looking good. Reformats should be done.

Response: We have checked 5 recent papers published in IJERPH and our tables conform with the tables in the published papers. If the Reviewer could provide specific suggestions for reformatting our tables, we would be happy to consider them.

  1. Conclusion in section 5 is also not strong. Please give a good summary and proper policy implications.

Response: Thank you for this suggestion. We have added a paragraph to the Conclusion that discusses recent initiatives that may help overcome some of the barriers to F2S participation experienced by small farmers (lines 370-381).

  1. It is known to all that every study suffers from some limitations, like data problem, model estimation/specification problem etc. But authors have not mentioned such limitations at the concluding section.

Response: The study limitations are given in the last paragraph of the Discussion.

  1. The paper should be revised in the light of above comments and suggestions before accepting for publication in the journal.

Response: Thank you for your time and effort in reviewing our work. We have revised the manuscript based on comments and suggestions from the Reviewer and other reviewers. We believe the revised manuscript is much improved and hope the revisions are acceptable to the Reviewer.

Reviewer 4 Report

The paper examined he problems in linking farmers to K-12 schools. Although introduction, methodology, analysis and conclusions are in line and informative. The study suffers from the small sample size.

The authors should have explored the data still more by using qualitative analysis as the sample size is limited. There is no qualitative information about specific problems faced by the different sections of the farmers categories. This type of information can add value to the paper.

Rather the authors completely depend on the econometric tools which are not quite suitable to understand the market linkages with the schools. Which type of commodities/value added products are more preferred by schools? Whether small farmers are having comparative advantage in selling to schools? Which type of produce getting more prices and remunerative? What are the food safety standards? Etc can be explored in the paper with more qualitative data.

Author Response

We thank the reviewers for their thoughtful and thorough review of our manuscript. We have incorporated many of the suggestions provided and believe that these revisions have improved the comprehensive nature and clarity of our manuscript. As is often the case, we had to balance incorporating reviewers’ suggestions with creating a comprehensive yet concise manuscript that will attract and hold readers’ attention and interest. We hope our responses and manuscript edits (made using track changes) have adequately addressed the reviewers’ concerns. Our responses to the comments and suggestions follow.

  1. The paper examined the problems in linking farmers to K-12 schools. Although introduction, methodology, analysis and conclusions are in line and informative. The study suffers from the small sample size.

Response: We thank the Reviewer for recognizing the strengths of our manuscript and agree that our sample size was small. We listed sample size as a limitation of our study in the last paragraph of the Discussion. We also have not attempted to generalize our study results to represent all small farmers in Mississippi or other regions in the US.

  1. The authors should have explored the data still more by using qualitative analysis as the sample size is limited. There is no qualitative information about specific problems faced by the different sections of the farmers categories. This type of information can add value to the paper.

Response: Thank you for bringing this to our attention. Because of the nature of our study (online survey) and to reduce participant burden, we did not collect qualitative information from the farmers who chose to complete the survey. We did collect information related to challenges or barriers that stop farmers from selling to K-12 schools which was reported in the Results and included in the Discussion. However, we can appreciate that challenges are a very important aspect that hinder connections between farmers and schools and hence have expanded upon the selling challenges reported by farmers in the current study (lines 265-280, 296-301).

  1. Rather the authors completely depend on the econometric tools which are not quite suitable to understand the market linkages with the schools. Which type of commodities/value added products are more preferred by schools? Whether small farmers are having comparative advantage in selling to schools? Which type of produce getting more prices and remunerative? What are the food safety standards? Etc can be explored in the paper with more qualitative data.

Response: Thank you for your questions. Because of the nature of our study (online survey) and to reduce participant burden, we did not collect qualitative information from the farmers who chose to complete the survey. We did address the Reviewer’s question related to commodities most preferred by schools (fruits and vegetables; lines 284-285]. We did not collect data from mid- or large-sized farmers and thus are unable to answer the question posed. We also cannot address the Reviewer’s question regarding which types of produce are most profitable for farmers to sell to schools as we did not collect such information. Such comparisons could likely only be made if several farmers were selling multiple types of produce to schools and even then, it may be farm/farmer specific. Finally, we have added a brief discussion and references pertaining to food safety standards (lines 276-280).

Round 2

Reviewer 1 Report

The considertions added in the discussion and conclusive sections have greatly improved the quality of the paper.

Author Response

  1. The considerations added in the discussion and conclusive sections have greatly improved the quality of the paper.

Response: We thank the Reviewer for recognizing the improvements in our manuscript.

Reviewer 2 Report

Literature Review

The authors should create a Section on Literature Review after the Introduction. This is important. We should have a global picture of the state of knowledge on this topic.

Materials and Methods

Please where is Delta State University? At least we should know the country. There could be more than one Delta State University in the World. Hence, the need to state the country, where this Delta State University is located.

Author Response

  1. Literature Review: The authors should create a Section on Literature Review after the Introduction. This is important. We should have a global picture of the state of knowledge on this topic.

Response: Thank you for this suggestion. We believe that an extensive F2S literature review is beyond the scope of our manuscript, would detract readers’ attention from the purpose of our study, and add unnecessary length to our manuscript. However, we can appreciate the Reviewer’s interest in such a review. Hence, we have added a sentence to the Introduction to point readers to a source for a 2021 comprehensive review of F2S in the United States (lines 49-51).  

  1. Materials and Methods: Please where is Delta State University? At least we should know the country. There could be more than one Delta State University in the World. Hence, the need to state the country, where this Delta State University is located.

Response: We have added the state and country location for the Delta State University IRB (line 80).

Reviewer 4 Report

Although, the paper suffer from a small sample size, they have examined the specific question of extent of Farm-to-School sales and limitations there in. Given that there is a limitation to use the qualitative data as the authors used online survey tools, they may extensively discussion their results by citing related literature across the world, how farmers are connected to markets especially to schools. What are the success stories. How schools can adjust like introducing online e-markets to buy directly from the nearby farmers. For this authors may refer Reddy, A. A. (2018). Electronic national agricultural markets. Current Science115(5), 826-837. and also Feenstra, G., & Ohmart, J. (2012). The evolution of the school food and farm to school movement in the United States: connecting childhood health, farms, and communities. Childhood Obesity (Formerly Obesity and Weight Management)8(4), 280-289.

and similar articles to enhance the scope of the paper.  

Author Response

  1. Although, the paper suffer from a small sample size, they have examined the specific question of extent of Farm-to-School sales and limitations there in. Given that there is a limitation to use the qualitative data as the authors used online survey tools, they may extensively discussion their results by citing related literature across the world, how farmers are connected to markets especially to schools. What are the success stories. How schools can adjust like introducing online e-markets to buy directly from the nearby farmers. For this authors may refer Reddy, A. A. (2018). Electronic national agricultural markets. Current Science, 115(5), 826-837. and also Feenstra, G., & Ohmart, J. (2012). The evolution of the school food and farm to school movement in the United States: connecting childhood health, farms, and communities. Childhood Obesity (Formerly Obesity and Weight Management), 8(4), 280-289. and similar articles to enhance the scope of the paper.

Response: Thank you for these suggestions. While a global review of F2S is certainly of interest, it is beyond the scope of our manuscript. We feel that it would detract from the results of our study as a reader may get the impression that we are attempting to generalize our study results to those of other regions in the US as well as other countries. Additionally, suggesting methods that schools may use to connect with farmers is also beyond the scope of our manuscript as the focus is on farmers and not schools. We have devoted a paragraph in the Discussion that provides methods that farmers may use to connect with schools (lines 296-311). Further, we have identified federal government policies and programs that may alleviate some of the challenges experienced by farmers when attempting to supply local food to schools (lines 356-368). Finally, we have added a sentence to the Introduction to point readers to a source for a 2021 comprehensive review of F2S in the United States (lines 49-51). Hence, we feel the scope of our manuscript is appropriate for the results we present.